# Tumor-Targeting Peptides Search Strategy for the Delivery of Therapeutic and Diagnostic Molecules to Tumor Cells

**DOI:** 10.3390/ijms22010314

**Published:** 2020-12-30

**Authors:** Maria D. Dmitrieva, Anna A. Voitova, Maya A. Dymova, Vladimir A. Richter, Elena V. Kuligina

**Affiliations:** The Institute of Chemical Biology and Fundamental Medicine of the Siberian Branch of the Russian Academy of Sciences, 630090 Novosibirsk, Russia; imaria819@gmail.com (M.D.D.); 2580_mana@mail.ru (A.A.V.); richter@niboch.nsc.ru (V.A.R.); kuligina@niboch.nsc.ru (E.V.K.)

**Keywords:** phage display, glioblastoma, tumor-targeting peptides, cancer stem cells (CSCs), CD133, CD44, immunocytochemistry, fluorescence-activated cell sorting (FACS)

## Abstract

Background: The combination of the unique properties of cancer cells makes it possible to find specific ligands that interact directly with the tumor, and to conduct targeted tumor therapy. Phage display is one of the most common methods for searching for specific ligands. Bacteriophages display peptides, and the peptides themselves can be used as targeting molecules for the delivery of diagnostic and therapeutic agents. Phage display can be performed both in vitro and in vivo. Moreover, it is possible to carry out the phage display on cells pre-enriched for a certain tumor marker, for example, CD44 and CD133. Methods: For this work we used several methods, such as phage display, sequencing, cell sorting, immunocytochemistry, phage titration. Results: We performed phage display using different screening systems (in vitro and in vivo), different phage libraries (Ph.D-7, Ph.D-12, Ph.D-C7C) on CD44+/CD133+ and without enrichment U-87 MG cells. The binding efficiency of bacteriophages displayed tumor-targeting peptides on U-87 MG cells was compared in vitro. We also conducted a comparative analysis in vivo of the specificity of the accumulation of selected bacteriophages in the tumor and in the control organs (liver, brain, kidney and lungs). Conclusions: The screening in vivo of linear phage peptide libraries for glioblastoma was the most effective strategy for obtaining tumor-targeting peptides providing targeted delivery of diagnostic and therapeutic agents to glioblastoma.

## 1. Introduction

Glioblastoma (GBM) is the most common and aggressive form of brain tumor, which is characterized by the least favorable prognosis—the average survival rate for patients with this diagnosis is 15 months [1]. In modern medical practice, standard methods such as surgery, radiation therapy and chemotherapy are used to treat glioblastoma, and in most cases these methods are ineffective. Such a low efficiency of glioblastoma treatment is often associated with two characteristic features of this tumor: the invasion of tumor cells into the brain parenchyma, which leads to the emergence of secondary tumor foci, and the high heterogeneity of tumor. A special contribution to the resistance of GBM cells to therapy is made by a small population of cells with a highly aggressive phenotype characteristic of cancer stem cells (CSCs) [2]. Targeted therapy based on the use of drugs specifically affecting specific types of tumors can be a solution to the problem of the low efficiency of the applied cancer therapies, which makes it possible to increase the effectiveness of treatment and minimize toxic effects on healthy tissues. The combination of the unique properties of cancer cells makes it possible to find specific ligands that interact directly with the tumor and ensure the implementation of the targeted approach. Currently, short peptides are considered promising agents for the delivery of therapeutic and diagnostic molecules to cancer cells, which have high affinity and specificity for the target and a higher efficiency of penetration into cancer cells as compared to ligands of larger sizes, for example, antibodies. One of the promising ways to search for tumor-targeting peptides is the screening of phage peptide libraries in tumor cell cultures in vitro and in xenograft models in vivo [3]. This approach can be applied to solve the problem of tumor heterogeneity, since the screening can reveal tumor-targeting peptides that specifically interact with different populations of tumor cells, including CSCs. A targeted approach to CSCs is especially relevant, since such characteristics of these cells as the ability to self-renewal, differentiation into various cell types, invasion of the brain parenchyma and metastasis, determine their resistance to chemotherapy and radiotherapy [2].

Earlier, by screening phage peptide libraries Ph.D-7 and Ph.D-12 (New England Biolabs, Ipswich, Massachusetts, USA), we selected bacteriophages displayed tumor-targeting peptides that provide specific binding of phage particles to human glioblastoma cells U-87 MG in vitro and with U-87 MG tumor in the xenograft model in vivo [4]. In this work, a screening of the Ph.D.-C7C phage peptide library was carried out to obtain tumor-targeting peptides to U-87 MG tumor cells with the phenotype of tumor stem cells (CD44+/CD133+), as well as a comparative analysis of the distribution in the body of mice and the specificity of the interaction with U87 MG tumor of bacteriophages displaying tumor-targeting peptides selected during biopanning of various peptide libraries in different selection systems.

## 2. Results

### 2.1. Biopanning of Linear Phage Libraries Ph.D.-12 and Ph.D.-7 on Cells and Tumors U-87 MG

Earlier, in our laboratory, we screened the phage peptide library Ph.D.-7 in vivo on U-87MG glioblastoma xenografts in immunodeficient mice. In the course of the work, 102 bacteriophages were selected; the sequences of 27 exposed peptides selected after the third round were identified and analyzed. When analyzing the sequences of the selected peptides, the highest frequency of occurrence was in the sequence HPSSGSA (92)—25.9% [4]. Additionally, the screening of the Ph.D.-12 phage peptide library in vitro on U-87 MG human glioblastoma cells was performed earlier. In the course of the work, 80 bacteriophages were selected; sequences of 39 exposed peptides selected after the third round and 37 peptides selected after the fifth round were identified and analyzed. After the fifth round, it was found that the sequence SWTFGVQFALQH (26) was found in 24.3% of cases [4].

### 2.2. Biopanning of the Circular Phage Peptide Library Ph.D.-C7C In Vivo and In Vitro

We carried out in vitro biopanning on cells of an immortalized human glioblastoma cell line U-87 MG using Ph.D.-C7C at the same protocol as for linear libraries. Three rounds of selection were carried out; the sequences of the exposed peptides providing the specific interaction of phage particles with U-87 MG cells were determined by sequencing. After the third round of biopanning, bacteriophages displayed the peptides PVPGSFQ (18C), PTQLHGT (23C), MHTQTPW (19C), TTKSSHS (2C), and ISYLYGR (36C) were selected. The frequency of occurrence of the peptides PVPGSFQ (18C) and PTQLHGT (23C) was 35% and 15%, respectively. Peptides MHTQTPW (19C), TTKSSHS (2C) and ISYLYGR (36C) accounted for 10% of the selected pool of bacteriophages (Figure 1).

### 2.3. Obtaining a Population of CD44+/CD133+ U-87 MG Cells for Selection of Bacteriophages Displaying Peptides Specific to CSCs

To obtain tumor-targeting peptides specific to U-87 MG cancer stem cells (CD44+/CD133+ cells), we screened the cyclic phage peptide library Ph.D.-C7C in vivo. The first two rounds of selection were performed on U-87 MG tumor transplanted subcutaneously into SCID mice. The third round of biopanning was performed on orthotopically implanted U-87 MG tumor into SCID mice. In this case, mice with a tumor were intravenously injected with an enriched phage peptide library after the first two rounds, after 24 h of circulation of the library in the body, the animals were euthanized and the tumor was removed. Tumor tissue was homogenized to single cells; tumor cells were stained for markers CD44, CD133 and sorted using Fluorescence-activated cell sorting (FACS). According to the results of sorting, the number of cells positive for CD44 (CD44+) was 8.9% (Figure 2A), positive for both markers (CD44+/CD133+)—5.53% (Figure 2B), positive for CD133 (CD44−/CD133+)—0.65% (Figure 2C).

Next, cells positive for both markers CD44/CD133 were lysed, the lysate was amplified in *Escherichia coli* and the sequence of the insert was determined by Sanger sequencing. According to the sequencing results, only one clone displaying the MHTQTPW peptide (No.19C) binds to cancer cells that were positive for both markers tested. It should be noted that the MHTQTPW peptide was previously selected in the biopanning on U-87 MG cells in vitro (data not shown).

### 2.4. Analysis of the Binding Specificity of Bacteriophages, Displaying Selected Peptides, to Human Glioblastoma Cells U-87 MG

We carried out a comparative analysis of the efficiency of binding of the bacteriophages displayed tumor-targeting peptides to human glioblastoma cells U-87 MG by fluorescence microscopy (Figure 3). We have previously shown that the peptide displayed by bacteriophage No. 26 ensures the binding and internalization of the phage particle into AS2 astrocytoma cells, but not into human MG1 glioblastoma cells [4]. In Figure 3 shows fluorescence microscopy of cells incubated with bacteriophages 19C, 36C, 92, 26, selected on different phage libraries, in different screening systems. Phage M13, displayed the peptide YTYDPWLIFPAN previously selected for MDA-MB 231 cells, was taken as a negative control [5]. No significant differences were found in the efficiency of binding to cells of bacteriophages displayed the studied peptides. Thus, the obtained tumor-targeting peptides are able to provide efficient specific binding of phage particles to U-87 MG glioblastoma cells.

### 2.5. Analysis of Biodistribution and Specificity of Accumulation of Bacteriophages, Displaying Selected Tumor-Targeting Peptides, in U87 MG Tumor Tissue

Comparative analysis of the distribution in the body of experimental animals and the specificity of accumulation in U-87 MG xenograft tumors of bacteriophages, displaying tumor-targeting peptides, was carried out by titration of tumor homogenates and tissues of control organs (kidney, liver, lungs, brain) after 4.5 h of circulation of phage particles in the body of the animal. For comparative analysis, bacteriophages No. 26 (Ph.D.-C12), No. 19c, No. 36c (Ph.D.-C7C) and No. 92 (Ph.D.-7) were selected. A random bacteriophage displayed peptide YTYDPWLIFPAN was used as a negative control. The titration data showed that bacteriophage No. 92, obtained by screening the phage peptide library Ph.D.-7 in vivo, accumulated to the greatest extent in the tumor tissue as compared to the control organs: the titer of the bacteriophage in the tumor exceeded its titer by more than 5.5 times in the kidneys, and more than 11 times in the brain, liver and lungs (Figure 4). 

Two-way analysis of variance (ANOVA) showed a statistically significant difference (*p* ≤ 0.0001) in the accumulation of this bacteriophage in the tumor as compared to the control phage and phages No. 26, No. 19C, No.36C. Bacteriophage No. 26 also specifically accumulated in the tumor tissue, but to a lesser extent compared to bacteriophage No. 92, its accumulation was statistically significantly different only from that for the control phage (*p* ≤ 0.001). Bacteriophages, selected from the cyclic library Ph.D.-C7C—No. 19C and No. 36C, showed the least accumulation in tumor tissue and other organs.

## 3. Discussion

The goal of this study was to develop a strategy for searching for tumor-targeting peptides for the delivery of therapeutic and diagnostic molecules to glioblastoma, which is characterized by some degree of heterogeneity. Tumor heterogeneity is due to small population of cells with a highly aggressive phenotype characteristic of CSCc. To identify CSCs, the level of CD24, CD29, CD44, CD133 and ALDH1 is most often examined. CD44 and CD133 are considered one of the most specific CSCs markers. CD44, a transmembrane glycoprotein, is considered one of the most important markers of CSCs [6]. As a result of alternative splicing, post-translational modifications, and partial cleavage by matrix metalloproteinases, multiple CD44 isoforms can exist in the cell [7]. CD44 acts as a co-receptor for several cell surface receptors (EGFR, Her2, Met6, TGFβRI, TGFβRII, VEGFR-2), thus participating in various signaling pathways (Rho, PI3K/Akt and Ras-Raf-MAPK), including those stimulating growth and cell motility. Another characteristic marker of CSCs, CD133 or prominin-1, is a transmembrane glycoprotein with a structure consisting of five transmembrane domains [8]. It is known that CD133 is required to maintain the properties of CSCs, and a low level of this marker in glioblastoma cells negatively effects on the ability of cells to self-renewal and neurosphere-forming [9]. The expression level of CD133 on cells is usually low, but can vary widely. Thus, in endometrial cancer, CD133 was immunohistochemically detected in 1.3–62.6% of cells, in colorectal cancer, CD133 was expressed in 0.3–82.0% of cells [10]. Despite the fact that CD133 is considered as a marker of CSCs, its studies as a marker of glioblastoma CSCs remain controversial [11]. Despite the unclear physiological function of CD133 in the pathogenesis of gliomas, mechanisms in which this receptor is involved have been discovered. It has been shown that under hypoxia an increase in the expression of this receptor is observed, as a result of which cells with a negative CD133 phenotype acquire a CD133+ phenotype [12]. Thus, at present, CSCs are considered the most promising targets for the search for specific therapeutic and diagnostic molecules. The use of combination therapy, including standard cytotoxic drugs capable of destroying the main tumor mass, and drugs targeting CSCs, can significantly increase the effectiveness of anticancer therapy and improve patient survival [13]. 

In this work, in order to develop a strategy for obtaining tumor-targeting peptides to glioblastoma, a comparative analysis of the binding efficiency of the selected peptides in the screening of linear and cyclic phage peptide libraries, Ph.D.-7, Ph.D.-12, and Ph.D.- C7C, in different selection systems (in vitro and in vivo) was conducted. We also used cyclic phage library, characterized by the fact that the peptides exposed on the surface protein p3 have a circular structure due to the formation of disulfide bridges between cysteines flanking the insert. It is believed that cyclic peptides are much less susceptible to proteolysis and often exhibit increased biological activity due to their conformational rigidity [14]. As a result of the studies carried out, it was found that all selected tumor-targeting peptides obtained from various peptide libraries, both in vitro and in vivo, are able to provide efficient specific binding of phage particles to not enriched U-87 MG glioblastoma cells. Indeed, the immunocytochemistry (Figure 3) showed that almost all cells in the population of not enriched U-87 MG cells are stained. On the image related to 19C phage, which was found after lysis the enriched cells (CD44+/CD133+) and their further amplification, not all cells were stained. One possible explanation of this fact could be that this peptide (19C) binds with some receptors of the stem cells surface which could not exist on all the cells in the general population, and likely with CD44 only, because according to cytometry data (Figure 2C), the population of CD44+/CD133+ cells is 5.53% only. Additionally, the phages No. 26, 92, 36C were found in the screening on unenriched U-87 MG cells. Another possibility is that after receiving the CD44+/CD133+ cells by sorting the CSCs could generate differentiated progeny, losing the markers of stemness. 

The highest specificity of binding to the xenograft U87 MG in vivo as compared to control organs is provided by linear tumor-targeting peptides obtained by screening the Ph.D.-12 phage peptide library on the xenograft U87 MG. Despite the great stability under physiological conditions and conformational rigidity, which often determines the high biological activity of cyclic peptides [14], the specificity of the interaction with the xenograft U-87 MG of bacteriophages displayed cyclic peptides selected on the population of glioblastoma cells expressing CSCs markers turned out to be lower than the specificity of interaction bacteriophages displaying linear peptides. Certain linear peptides are believed to have conformation recognized by target receptors without the need for cyclization. In addition, the linear conformation of the peptide can provide a greater efficiency of its penetration into the cell as compared to cyclic peptides, since a large free energy is required for penetration into the cell [15]. Additionally, when studying the distribution and binding of phage particles to a tumor xenograft, we must take into account the fact that the number of CD44+/CD133+ cells inside xenograft is small. In addition to U-87 MG cells, there are the endothelial cell and stroma’s cells in the tumor. So, CSCs will be in small quantities in the tumor tissue, which explains the absence of significant differences between the binding of the control phage and bacteriophage No. 19C to the U87 MG xenograft. So, using the strategy of searching for peptides on population of enriched cells using specific markers (CD44+/CD133+), we met some obstacles in further experiments. Thus, according to the totality of the obtained data, the most effective strategy for obtaining tumor-targeting peptides that provide targeted delivery of diagnostic agents and therapeutic drugs to human glioblastoma tumors is to screen linear phage peptide libraries for glioblastoma tumors in vivo.

## 4. Materials and Methods

### 4.1. Cell Cultures

Cancer cell line U-87 MG was obtained from the Russian cell culture collection (Russian Branch of the ETCS, St. Petersburg, Russia). U-87 MG cells were cultivated in alpha-MEM (Thermo Fisher Scientific, Waltham, MA, USA) supplemented with 10% of fetal bovine serum (FBS) (Sigma, St. Louis, MO, USA), 1 mM L-glutamine, 250 mg/mL amphotericin B and 100 U/mL penicillin/streptomycin. Cells were grown in a humidified 5% CO2–air atmosphere at 37 °C and were passaged with TripLE Express Enzyme (Thermo Fisher Scientific, USA) every 3–4 days. 

### 4.2. Animals

Female SCID hairless outbred (SHO-Prkdc^scid^Hrhr) mice aged 6–8 weeks were obtained from «SPF-vivarium» ICG SB RAS (Novosibirsk, Russia). Mice were housed in individually ventilated cages (Animal Care Systems, Centennial, Colorado, USA) in groups of one to four animals per cage with ad libitum food (ssniff Spezialdiäten GmbH, Soest, Germany) and water. Mice were kept in the same room within a specific pathogen-free animal facility with a regular 14/10 h light/dark cycle (lights on at 02:00 h) at a constant room temperature of 22 ± 2 °C and relative humidity of approximately 45 ± 15%. 

### 4.3. In Vivo and In Vitro Biopanning 

Biopanning of the phage peptide library (Ph.D.-C7C, New England Biolabs, Ipswich, MA, USA) on U-87 MG glioblastoma cells in vitro was performed as described previously with some modifications [16,17], namely. The cells that reached 100% confluence were washed with 4 mL of PBS, then 400 μL of 10 mM EDTA was added to detach the cells from the surface and incubated for 4 min at 37 °C. Then 1 mL of complete growth medium was added and cell suspension was transferred into a falcon with a volume 15 mL. The cells were centrifuged for 3 min at 1000 rpm, the supernatant was removed, the cells were resuspended in 4 mL of PBS, and the centrifugation was repeated. The cells were resuspended in 4 mL of blocking buffer (5% BSA/PBS), incubated for 10 min at 37 °C and centrifuged for 3 min at 1000 rpm. The supernatant was removed, the cells were washed with 4 mL PBS and pelleted by centrifugation (3 min, 1000 rpm). The supernatant was removed, the cells were incubated with 3 mL of a negative selection-depleted phage peptide library for 1 h at 4 °C and centrifuged for 3 min at 1000 rpm. The supernatant was removed, the cell pellet was washed three times with 4 mL of PBS and centrifuged for 3 min at 1000 rpm. The cells were resuspended in 4 mL of growth medium heated to 37 °C to provide conditions for the internalization of bacteriophages into cells, incubated for 15 min at 37 °C and centrifuged for 3 min at 1000 rpm. The cells were then washed three times with 4 mL of PBS. 400 μL of Triple Express was added to the cell pellet to remove non-internalized bacteriophages, incubated for 2 min at 37 °C, 1 mL of complete growth medium was added, and centrifuged for 3 min at 1000 rpm. The supernatant was removed, the cells were washed with 4 mL PBS, and the centrifugation was repeated. Then, the cells were lysed with 1 mL of mQ water for 20 min at room temperature. The cell lysate was centrifuged for 5 min at 14,000 rpm, the supernatant was removed, and the phage suspension (1 mL) was amplified. The amplified population of phage particles was used for subsequent rounds of selection. 

For in vivo screening, we used the previously described methods [5,18], to wit. SCID mice with subcutaneously and orthotopic glioblastoma xenograft U-87 MG were injected into the tail vein with 300 μL of a phage peptide library with a concentration of 2 × 10^11^ pfu/mL, diluted in saline. The circulation time of the phage library in the bloodstream for mice with subcutaneously glioblastoma xenograft U-87 MG was 5 min; for mice with orthotopic glioblastoma xenograft U-87 MG, the circulation time was 24 h. After the screening time elapsed, the mouse was sacrificed by cervical dislocation, the chest was opened, and 15 mL of saline was perfused through the heart to remove bacteriophages which not binding with the tumor from the bloodstream. The tumor was removed, washed in saline and homogenized in 1 mL PBS containing 1 mM PMSF. The tumor tissue homogenate was centrifuged for 10 min at 10,000 rpm. The pellet was resuspended in 1 mL of blocking buffer (1% BSA), after which centrifugation was repeated under the same conditions. The pellet was resuspended in 1 mL of liquid culture of *E. coli* ER2738 in the average log-phase with an optical density 0.3 (OD600) for elution of bacteriophages bound to the tumor and incubated for 30 min at 37 °C at 170 rpm. The eluate of phage particles was centrifuged for 5 min at 10,000 rpm. The supernatant was transferred to separate tubes and the enriched phage library was amplified for subsequent rounds of selection. 

Manipulations on glioblastoma xenograft U-87 MG and monitoring of tumor growth were carried out by employees of «SPF-vivarium» ICG SB RAS. After the third round of selection, phage particles were titrated to obtain individual phage colonies, which were used for DNA isolation according to the manufacturer’s protocol for the phage display peptide library. The sequencing reaction products were determined using an ABI 310 Genetic Analyzer (Applied Biosystems, Foster City, CA, USA) at the Genomics Core Facility of SB RAS using sequencing primers (-96III (5′-CCC TCA TAG TTA GCG TAA CG-3′)).

### 4.4. Tumor Preparation for Cell Sorting

In mice with orthotopical glioblastoma xenograft U-87 MG, a peptide library enriched with in vivo biopanning (2 × 10^11^ PFU/mL of phage particles in 500 μL of saline) was injected into the tail vein. After 24 h, the mouse was sacrificed by cervical dislocation and the tumor was removed. The tumor was washed twice with PBS containing 10% penicillin-streptomycin (Sigma-Aldrich, St. Louis, MO, USA), after which it was crushed with a scalpel on a Petri dish, transferred into a falcon with 3 mL of trypsin and incubated in a water bath at 37 °C for 10 min to dissociate the cells. To inactivate trypsin, 3 mL of a trypsin inhibitor from soybeans (Sigma-Aldrich, USA) was added to the cell suspension, after which the cells were centrifuged for 10 min at 800 rpm. The cell pellet was resuspended in NSC medium for neural stem cells (Sigma-Aldrich) until a homogeneous cell suspension was formed. The undissociated pieces of tumor tissue were removed and additionally homogenized. 10 mL of NSC medium was added to the cell suspension, filtered through a filter with a pore size of 40 μm, and centrifuged for 10 min at 800 rpm. The cells were resuspended in 1 mL of NSC medium and incubated for 2 h at 37 °C to restore the proteomic profile of the cells. 

### 4.5. Cell Sorting

After incubation in NSC medium, cells were incubated in 500 μL blocking buffer containing 10% FBS for 10 min. The cells were then washed with 500 μL PBS and incubated for 45 min on ice with primary antibodies against CD44 labeled with FITC (Abcam, Cambridge, UK) and primary antibodies against CD133 labeled with Alexa Fluor 647 (Abcam), both diluted in 1% FBS in PBS, in 200 μL. The cells were washed twice with 500 μL PBS, resuspended in 500 μL PBS containing 4 μg/mL gentamicin (Thermo Fisher Scientific, Waltham, MA, USA) and passed through a strainer (BD Biosciences, Franklin Lakes, NJ, USA) into flow cytometry tubes (BD Biosciences). The analysis and sorting of cells was carried out on a SONY SH800S Cell Sorter (Sony Biotechnology, San Jose, CA, USA).

### 4.6. Immunocytochemistry 

U-87 MG cells were incubated on BD Falcon culture slides to 80–90% confluence, washed with PBS twice, and 100μL of the selected phage clone (2 × 10^10^ PFU/mL) in PBS-BSA Ca/Mg buffer (0.1% BSA, 1mM CaCl_2_, 10 mM MgCl_2_ × 6H_2_O); was added. Cells were incubated with the bacteriophage clone for 2 h at 37 °C with the following treatment according to the previously described technique with slight modifications [5], namely. After incubation at 37 °C, cells were washed three times with 500 μL buffer (100 mM glycine, 0.5 M NaCl, pH 2.5) at room temperature, fixed with 200 μL cold 4% formaldehyde for 10 min and washed twice with PBS. Then, 200 μL 0.2% Triton X100 was added for 10 min to permeabilize cells, after which the cells were washed twice with 500 μL PBS. Next, cells were incubated with 200 μL mouse Anti-M13 Bacteriophage Coat Protein g8p antibodies (Abcam) diluted in 1% BSA/PBS buffer (1:200) for 45 min at 4 °C and washed four times with cold 500 μL 1% BSA/PBS buffer. Next, cells were incubated with 200 μL secondary Alexa Fluor 647 (Abcam, UK) diluted in 1% BSA/PBS buffer (1:200) for 45 min at 4 °C and washed four times with 500 μL cold 1% BSA/PBS buffer. Then the cells were stained with DAPI (Thermo Fisher Scientific) and analyzed by fluorescent microscopy Axio Skope 2 Plus (Zeiss, Oberkochen, Germany) at the Center for Microscopic Analysis of Biological Objects of SB RAS (Novosibirsk, Russia).

### 4.7. Analysis of the Specificity of Accumulation of Bacteriophages Displayed Selected Peptides in Glioblastoma Xenograft U-87 Mg

Mice with a subcutaneously transplanted tumor were injected into the tail vein with 500 μL of bacteriophage (2 × 10^9^ PFU/mL) diluted in physiological solution. After 4.5 h of circulation of phage particles in the body, the mouse was sacrificed by cervical dislocation and perfused through the left ventricle of the heart with 15 mL of saline. Then the tumor and control organs (liver, kidney, lungs, and brain) were removed, washed in PBS, and homogenized in 1 mL PBS containing 1 mM PMSF (Sigma Aldrich). The homogenates of tumor tissue and control organs were centrifuged for 20 min at 10,000 g at room temperature to elute bound bacteriophages and were resuspended. The resulting suspension of phage particles was titrated on agar LB medium supplemented with 1 mg/mL X-Gal and 1.25 mg/mL IPTG.

### 4.8. Statistical Analysis

Two-way ANOVA was used for comparisons of more than two sets of data. Differences were considered to be significant if the *p*-value was <0.05. Nucleotide sequences of the inserts encoding peptides were analyzed using MEGA X software.

## Figures and Tables

**Figure 1 ijms-22-00314-f001:**
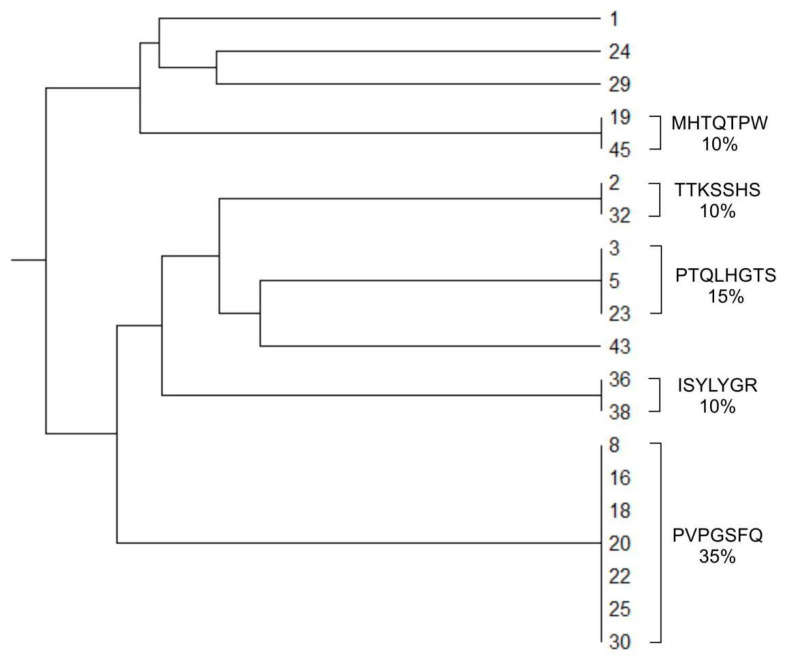
Phylogenetic tree built on the basis of nucleotide sequences of peptides displayed by bacteriophages obtained by screening the Ph.D.-C7C library in vitro on U-87 MG cells, using the Unweighted Pair Group Method with Arithmetic Mean (UPGMA method) and the MEGA X software.

**Figure 2 ijms-22-00314-f002:**
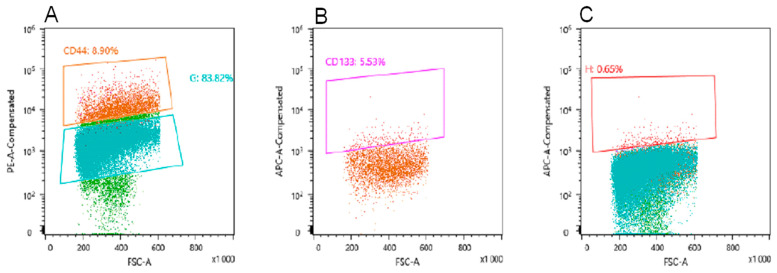
Evaluation of CSCs markers CD44 and CD133 using FACS-analysis. (**A**)—population of cells, positive for CD44 (orange); (**B**)—population of CD44-positive cells with a population of CD133-positive cells detected in it (pink); (**C**)—population of cells positive for the marker CD133 (red).

**Figure 3 ijms-22-00314-f003:**
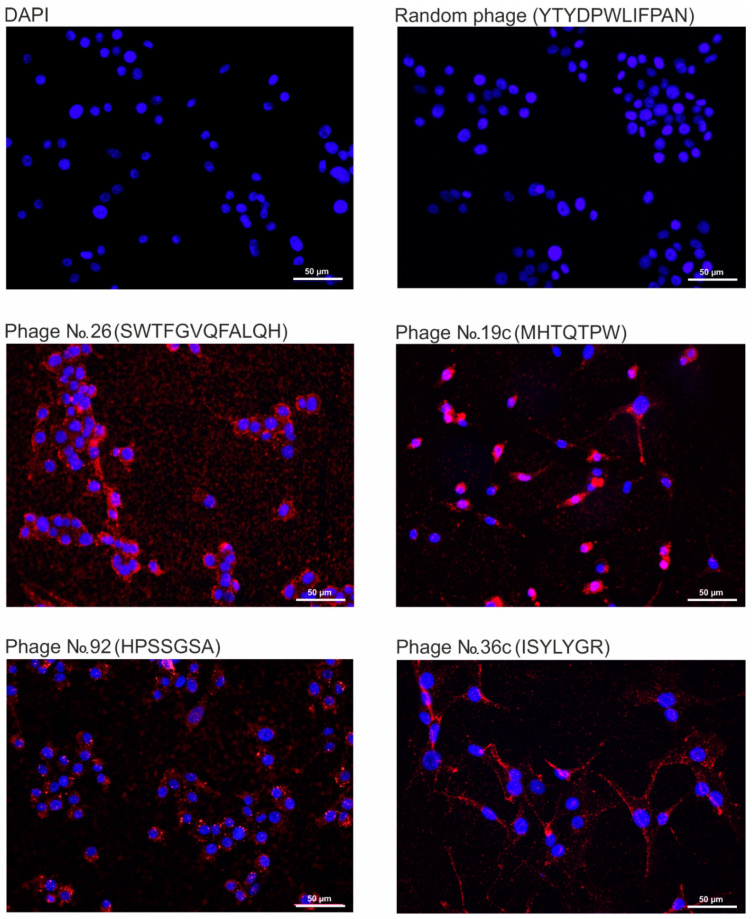
Fluorescence microscopy of U-87 MG cells incubated with bacteriophages No. 26, 19C, 92, 36C, displayed the peptides SWTFGVQFALQH, MHTQTPW, HPSSGSA, ISYLYGR, respectively. Microscopy was performed using anti-M13 g8p monoclonal antibody and anti-Mouse IgG (H + L) Alexa Fluor 647. DAPI was used to visualize cell nuclei. DAPI—DAPI stained U-87 MG cells not incubated with phage.

**Figure 4 ijms-22-00314-f004:**
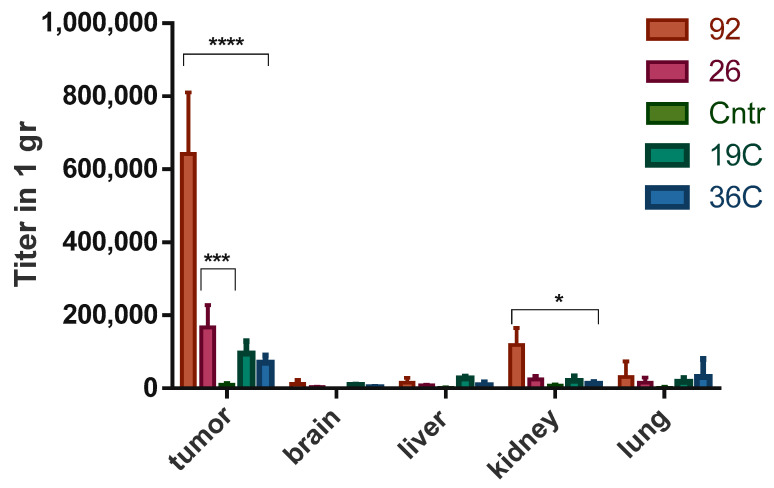
Comparative analysis of the distribution and specificity of accumulation in the U-87 MG xenograft Table 92, 26, 19C and 36C displayed peptides HPSSGSA, SWTFGVQFALQH, MHTQTPW, ISYLYGR, respectively, and the control bacteriophage displaying the peptide YTYDPWLIFPAN (control). The average titer of bacteriophages (pfu/1 g of tissue) obtained from the tumor and control organs (liver, brain, kidney and lungs). Data are presented as mean ± standard deviation. Multiple comparison was performed using two-way ANOVA. * *p* ≤ 0.05; *** *p* ≤ 0.001; **** *p* ≤ 0.0001.

## Data Availability

Data is contained within the article.

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
