# Peer review of "Tumor-Targeting Peptides Search Strategy for the Delivery of Therapeutic and Diagnostic Molecules to Tumor Cells"

_ijms, 2020, doi:10.3390/ijms22010314_

Round 1

Reviewer 1 Report

The submitted manuscript describes screening for targeting peptides for glioblastoma treatment. As illustrated in the introduction glioblastoma stem cells as the most therapy-resistant population of this aggressive brain tumor should be targeted. Targeting of cancer stem cells is a good approach to deliver anti-cancer agents but selectivity for the targeting by the selected peptides is not clearly shown in the study.

Comments

- Instead of citation of previous references, more information in the description of the methods should be given.

-Based on the motivation as illustrated in the Introduction, it appears that the selected peptides should bind predominantly to CD44+/CD133+ cells. However, Fig. 3 shows a heterogeneous population of presumably not enriched U-87 MG cells, where almost all cells are stained. To demonstrate selectivity of the targeting, staining of enriched and not enriched cells should be shown.

-It would also be useful to compare binding to a normal human glial cell line or other human cells to exclude general differences between murine and human cells.

Minor

Mention the ethical approval number for the animal experiments.

Author Response

We thank reviewer for a thorough revision of our manuscript. Please find below our response to reviewer’s comments.

Point 1: Instead of citation of previous references, more information in the description of the methods should be given.

Response 1: Corrected. We have expanded the section «Materials and methods, we highlighted (red highlighted text) this new details in the section.

Point 2: -Based on the motivation as illustrated in the Introduction, it appears that the selected peptides should bind predominantly to CD44+/CD133+ cells. However, Fig. 3 shows a heterogeneous population of presumably not enriched U-87 MG cells, where almost all cells are stained. To demonstrate selectivity of the targeting, staining of enriched and not enriched cells should be shown.

Response 2: Indeed, Figure 3 shows a population of not enriched U-87 MG cells, where almost all cells are stained. On the picture related to 19C phage which was found after lysis the enriched cells (CD44+/CD133+) and further amplification not all cells were stained. Possible explanation of this fact could be that this peptide (19C) binds with some receptors of the stem cells surface which could be exist on not all cells in the general population, and likely with CD44 only, because according to cytometry data (Fig.2, C), the population of CD44+/CD133+ cells is 0,65% only.

Also the phages No. 26, 92, 36C were found in the screening on not enriched U-87 MG cells. Another point it is that after receiving the CD44+/CD133+ cells by sorting (and it was very rare population – 0,65%) this CSCs could generate differentiated progeny, with losing the markers of stemness. With each passage, it will be difficulty and controversially to compare «enriched» and not enriched cells for these markers.

Also, when studying the distribution and binding of phage particles to a tumor xenograft, we must take into account the fact that the number of CD44 + / CD133 + cells inside xenograft is small. In addition to U-87 MG cells, there are the endothelial cell and stroma’s cells in the tumor. So, stem cancer cells will be in small quantities in the tumor tissue, which explains the absence of significant differences between the binding of the control phage and bacteriophage No. 19C to the tumors.

That is why we decided only to demonstrate selectivity of the targeting, staining of not enriched cells.

Point 3: It would also be useful to compare binding to a normal human glial cell line or other human cells to exclude general differences between murine and human cells.

Response 3: We understand that it would be better to compare binding to a normal human glial cell line or other human cells to exclude general differences between murine and human cells. There are also several points on which we can consider that our model is suitable at this stage. In our experiments we were guided by «NONCLINICAL EVALUATION FOR ANTICANCER PHARMACEUTICALS», S9, 4 version, dated 29 October 2009, where for nonclinical studies could be used animal species to evaluate the distribution of the drug. We have found these tumor–targeting peptides for further conjugation with the candidate antitumor drugs. In future we will plan to evaluate the binding specificity of the received drug (tumor-targeting peptide conjugated with anticancer agent) with different cell lines. Also we understand that tumors in SCID mice cannot faithfully copy the microenvironment of human tumors, which makes these models unsuitable, for example, for immunotherapy. Therefore, we could use the humanized PDX (Hu-PDX) model. But, humanized mouse models are rather expensive, time consuming, and require integration of multidisciplinary expertise (Meraz et al., 2019).

Point 4: Mention the ethical approval number for the animal experiments.

Response 4: Corrected. We added the ethical approval number for the animal experiments the relevant section: «The protocol was approved by Inter-Institutional Bioethics Commission at the Institute of Cytology and Genetics SB RAS (Permit Number: 68, Date: 01.12.2020) ».

Corrected text is displayed in red color.

We highlighted (red highlighted text) all changes made when revising the manuscript to make it easier for the Editors to give a prompt decision on manuscript.

Reviewer 2 Report

The work conducted by Dmitrieva et al. provides a screening of the Ph.D.-C7C phage peptide library with the goal of obtaining tumor-targeting peptides to U-87 MG tumor cells with the phenotype of tumor stem cells (CD44 + / CD133 +). It also offers a comparative analysis of its distribution and the specificity of the interaction formed with the phages.

The work is very well organized, with the information being presented in a precise and concise manner. It was a very quick and enjoyable reading, without distracting from the goal. The data is compelling and the discussion is scientifically sound. 

The authors did a good job and as such I recommend the publication of this manuscript in its present form.

Author Response

We thank reviewer for a thorough revision of our manuscript.

Round 2

Reviewer 1 Report

I thank the authors for their replies and agree with the argumentation. The only remaining point in my view is that l.59-63 give the impression that CSC-specific targeting is the goal and in the manuscript the explanation, which is given very well in the reply, is missing why all tumor cells are stained.

Author Response

We thank again reviewer for a thorough revision of our manuscript. Please find below our response to reviewer’s comment.

Point 1: The only remaining point in my view is that l.59-63 give the impression that CSC-specific targeting is the goal and in the manuscript the explanation, which is given very well in the reply, is missing why all tumor cells are stained.

Response 1: Corrected. We added the text to the section «Discussion» (l. 192-202, l.213-220), where the readers can find the answer why we took not enriched U-87 MG cell line for ICC. And the strategy of searching for peptides on population of enriched cells using specific marker can meet obstacles in further experiments.  

We highlighted (red highlighted text) all changes made when revising the manuscript to make it easier for the Editors to give a prompt decision on manuscript.